# Clinical Outcomes of a Zika Virus Mother–Child Pair Cohort in Spain

**DOI:** 10.3390/pathogens9050352

**Published:** 2020-05-07

**Authors:** Antoni Soriano-Arandes, Marie Antoinette Frick, Milagros García López-Hortelano, Elena Sulleiro, Carlota Rodó, María Paz Sánchez-Seco, Marta Cabrera-Lafuente, Anna Suy, María De la Calle, Mar Santos, Eugenia Antolin, María del Carmen Viñuela, María Espiau, Ainara Salazar, Borja Guarch-Ibáñez, Ana Vázquez, Juan Navarro-Morón, José-Tomás Ramos-Amador, Andrea Martin-Nalda, Eva Dueñas, Daniel Blázquez-Gamero, Resurrección Reques-Cosme, Iciar Olabarrieta, Luis Prieto, Fernando De Ory, Claire Thorne, Thomas Byrne, Anthony E. Ades, Elisa Ruiz-Burga, Carlo Giaquinto, María José Mellado-Peña, Alfredo García-Alix, Elena Carreras, Pere Soler-Palacín

**Affiliations:** 1Paediatric Infectious Diseases and Immunodeficiencies Unit, Hospital Universitari Vall d’Hebron, Vall d’Hebron Research Institute, Universitat Autònoma de Barcelona, 08035 Barcelona, Spain; mafrick@vhebron.net (M.A.F.); mespiau@vhebron.net (M.E.); salazarainara2@gmail.com (A.S.); andmartin@vhebron.net (A.M.-N.); psoler@vhebron.net (P.S.-P.); 2Department of Paediatrics, Hospital Universitario La Paz, 28046 Madrid, Spain; mghortelano@salud.madrid.org (M.G.L.-H.); mariajose.mellado@salud.madrid.org (M.J.M.-P.); 3Departament of Microbiology, Hospital Universitari Vall d’Hebron, 08035 Barcelona, Spain; esulleir@vhebron.net; 4Unit of Fetal Medicine, Department of Obstetrics, Hospital Universitari Vall d’Hebron, 08035 Barcelona, Spain; crodo@vhebron.net (C.R.); asuy@vhebron.net (A.S.); ecarrera@vhebron.net (E.C.); 5Instituto de Salud Carlos III, 28029 Madrid, Spain; paz.sanchez@isciii.es (M.P.S.-S.); a.vazquez@isciii.es (A.V.); fory@isciii.es (F.D.O.); 6Department of Neonatology, Hospital Universitario La Paz, 28046 Madrid, Spain; mcabreral@salud.madrid.org; 7Department of Obstetrics, Hospital Universitario La Paz, 28046 Madrid, Spain; mcallefm@gmail.com (M.D.l.C.); eantolin@salud.madrid.org (E.A.); 8Department of Paediatrics, Hospital General Universitario Gregorio Marañón, 28009 Madrid, Spain; marimarsantos5@gmail.com (M.S.); evadue9@gmail.com (E.D.); 9Department of Obstetrics, Hospital General Universitario Gregorio Marañón, 28009 Madrid, Spain; mcvinuela73@hotmail.com; 10Department of Paediatrics, Hospital Universitari Josep Trueta, 17007 Girona, Spain; borjaguarch@hotmail.com; 11Hospital Costa del Sol, 29603 Marbella, Spain; jnavarromoron@gmail.com; 12Department of Paediatrics, Hospital Universitario Clínico San Carlos, 28040 Madrid, Spain; josetomas.ramos@salud.madrid.org; 13Pediatric Infectious Diseases Unit, Hospital Universitario 12 de Octubre, Madrid, Instituto de Investigación Hospital 12 de Octubre (imas12), Universidad Complutense, 28041 Madrid, Spain; danielblazquez@hotmail.com (D.B.-G.); luismanuel.prieto@salud.madrid.org (L.P.); 14Hospital El Escorial, 28200 Madrid, Spain; resureques@hotmail.com; 15Hospital Severo Ochoa, Leganés, 28911 Madrid, Spain; iciar.olabarrieta@gmail.com; 16University College London Great Ormond Street Institute of Child Health, GOSH NIHR BRC, London WC1N 1EH, UK; claire.thorne@ucl.ac.uk (C.T.); t.byrne@ucl.ac.uk (T.B.); e.burga@ucl.ac.uk (E.R.-B.); 17University of Bristol, Bristol BS9 1AF, UK; t.ades@bristol.ac.uk; 18Division of Pediatric Infectious Diseases, Department for Woman and Child Health, University of Padova, 35122 Padova, Italy; carlo.giaquinto@unipd.it; 19Fundació NeNe, 28010 Madrid, Spain; alfredoalix@gmail.com

**Keywords:** zika virus, microcephaly, congenital infection, adverse outcome, arboviruses

## Abstract

Background: Zika virus (ZIKV) infection has been associated with congenital microcephaly and other neurodevelopmental abnormalities. There is little published research on the effect of maternal ZIKV infection in a non-endemic European region. We aimed to describe the outcomes of pregnant travelers diagnosed as ZIKV-infected in Spain, and their exposed children. Methods: This prospective observational cohort study of nine referral hospitals enrolled pregnant women (PW) who travelled to endemic areas during their pregnancy or the two previous months, or those whose sexual partners visited endemic areas in the previous 6 months. Infants of ZIKV-infected mothers were followed for about two years. Results: ZIKV infection was diagnosed in 163 PW; 112 (70%) were asymptomatic and 24 (14.7%) were confirmed cases. Among 143 infants, 14 (9.8%) had adverse outcomes during follow-up; three had a congenital Zika syndrome (CZS), and 11 other potential Zika-related outcomes. The overall incidence of CZS was 2.1% (95%CI: 0.4–6.0%), but among infants born to ZIKV-confirmed mothers, this increased to 15.8% (95%CI: 3.4–39.6%). Conclusions: A nearly 10% overall risk of neurologic and hearing adverse outcomes was found in ZIKV-exposed children born to a ZIKV-infected traveler PW. Longer-term follow-up of these children is needed to assess whether there are any later-onset manifestations.

## 1. Introduction

Zika virus (ZIKV) infection constitutes one of the most challenging infectious epidemic outbreaks worldwide to date. ZIKV is a uniquely challenging emerging zoonotic pathogen, with multiple routes of transmission and teratogenic potential in fetuses [1]. The association of ZIKV infection in pregnancy with congenital microcephaly and other neurodevelopment abnormalities in fetuses and infants was a crucial factor underscoring the public health importance of this emergent virus [2,3,4,5,6]. In response to the epidemic, the World Health Organization (WHO) recognized ZIKV as a public health emergency of international concern (PHEIC) on 1 February 2016 [7]. 

Prenatal exposure to ZIKV is now a well-documented cause of severe microcephaly in infants [8]. In addition, evidence indicates that clinical presentation of congenital ZIKV syndrome (CZS) differs in severity and prognosis depending on the head circumference (HC) at birth and the central nervous system (CNS) damage [8]. Adverse clinical outcomes at birth or over the first year of life have been described in different cohorts, affecting between 10% and 45% of the offspring exposed to ZIKV in utero [9,10,11,12,13,14]. Most of these infants are expected to survive, and their estimated life span is expected to be comparable to other children with microcephaly, epilepsy, and intellectual disability [15]. It is important to note that consensus is lacking about the phenotype of CZS; however, fetal brain disruption sequence [4,16,17] and neuropathology resembling congenital infections, such as toxoplasmosis or cytomegalovirus (CMV), have been reported among infants with CZS [18]. Therefore, the long-term impact of CZS may be severe for children and their families. The literature to date on CZS and children exposed to ZIKV in utero is dominated by studies conducted in epidemic and endemic settings, where individuals may be (re-)exposed to other flavivirus.

Spain started to report cases of Zika from 2016. Consequently, the government implemented ZIKV surveillance of all pregnancies, as part of antenatal and maternal care, including women coming from or having travelled to epidemic regions. Here, we report findings from the PedZikaRed study, which was set up in Spain to take advantage of this unique opportunity to study ZIKV infection during pregnancy and infant outcomes in a setting where ZIKV was not circulating. 

## 2. Results

### 2.1. Epidemiological and Clinical Characteristics of the Participants

A total of 179 pregnant women with ZIKV suspected infection were enrolled from January 2016 to February 2019. Participants were classified as a confirmed or probable ZIKV case following the national guidelines [19]. A confirmed case was defined as a pregnant woman with (i) a positive reverse transcriptase polymerase chain reaction (RT-PCR) test for ZIKV in serum and/or urine samples and/or (ii) with a positive IgM against ZIKV, a negative IgM against dengue virus (DENV), and a ZIKV neutralization titer ≥ 1/32. A ZIKV probable case was defined as a pregnant woman with an isolated positive IgG test against ZIKV and a ZIKV neutralization titer ≥1/32 (Table 1).

For the analysis, 16 were excluded because of empty (n = 3) or duplicate records (n = 2), missing date of diagnosis (n = 1), retrospective testing (n = 2), infection during a previous pregnancy (n = 1), or not meeting eligibility criteria (i.e., no indication of infection) (n = 7). Thus, this paper presents the data from 163 pregnant women and their infants included in the mother–child pair analyses (Figure 1). 

The vast majority were Latin American immigrants (95.7%) who had travelled to visit family and friends, whereas a small group was represented by European-born women (7/163; 4.3%) who travelled to Latin America for tourism (Table 2). 

### 2.2. Laboratory Confirmation of Maternal Infection

Screening for ZIKV occurred at a median (IQR) of 21 (14–30) gestational weeks. Median [IQR] time of mother’s exposure at risk for ZIKV during pregnancy was 86 (31–162) days. Twenty-four pregnant women (24/163; 14.7%) were classified as having confirmed ZIKV infection, of whom 14 tested positive for ZIKV RT-PCR, and ten tested positive for ZIKV-IgM and the ZIKV plaque reduction neutralization test (PRNT). The rest of the participants were classified as probable maternal ZIKV infection (139/163; 85.3%). From the whole group, 70% were asymptomatic, and only 21% of women with probable ZIKV infection had symptoms compared to 83% of the 24 women with confirmed ZIKV infection (p<0.0001). Rash was the most prevalent clinical finding (78%), followed by fever (67%) and arthralgia (64%). Amniocentesis was performed in 30 women (19%), mostly in the group with confirmed infections (16/24; 70%). Amniotic fluid ZIKV RT-PCR was positive in two of the cases with CZS, the rest tested negative. Additionally, among 114 women screened for DENV infection, 109 tested positive for DENV-IgG, and eight also tested positive for DENV-IgM during pregnancy. From the 72 women screened for chikungunya virus (CHKV), 27 tested positive for CHKV-IgG, and three of these women had a positive CHKV-IgM result. One woman was found positive for human immunodeficiency virus (HIV) as well as ZIKV, but had an HIV-uninfected infant who was healthy at birth and throughout follow-up. No other active infections were detected among participants.

### 2.3. Ultrasound and MRI Results during Pregnancies

Prenatal ultrasound was performed at least once in all but three women (160/163; 98.2%). Second trimester ultrasound was normal in all but five cases (three with brain malformations associated with CZS, one congenital heart disease associated with a 22q11 deletion syndrome, and one termination of pregnancy (TOP) due to CZS). Neurologic magnetic resonance imaging (MRI) was performed on 43 (26.4%) pregnant women; three showed fetal CNS abnormalities (3/43; 7%): two of the CZS cases, and one infant with proportionate microcephaly born at 40 weeks of gestation that was lost-to-follow-up after first postnatal visit.

### 2.4. Maternal, Fetal, and Neonatal Outcomes

This section details the results based on 142 women who delivered 143 live-birth children (Table 3). The median (IQR) duration of follow-up of the infants was 31 (IQR 13–50) weeks, with a maximum follow-up time of 798 days. During the follow-up, 9.8% (14/143) (95%CI: 5.5–15.9%) infants had adverse outcomes. There were three cases of CZS (two live-birth, one stillbirth), all born to mothers with ZIKV-confirmed infection. The overall incidence of CZS was 2.1% (95%CI: 0.4–6.0%), and 15.8% (3/19) (95%CI: 3.4–39.6%) among ZIKV-confirmed mothers. Eleven children had other potentially Zika-related outcomes (OPZRO): five (3.5%) were diagnosed with mild hearing loss using a sequence of automated auditory brainstem response (AABR) tests (one subsequently diagnosed with language impairment at 24 months of age) and six infants had abnormal neuroimaging results using either MRI or ultrasound CNS scans (Table 4). Other congenital infections were ruled out for all cases, including congenital CMV infection through viral load test in urine or saliva. None of the children with OPZRO presented with any abnormal neurological development after birth. None of the three CZS cases tested positive for ZIKV-IgM or ZIKV RT-PCR (blood, urine, CSF, and saliva) in the postnatal period. ZIKV-IgG was initially positive for the two live-born CZS cases, but seroreversion was observed in both after the first 12 months of life. Of note, the 22q11 deletion syndrome case detected was excluded from the OPZRO group because there is no evidence to associate this pathology with ZIKV infection among children.

## 3. Discussion

To our knowledge, this is the first prospective cohort study of mothers and their children exposed to ZIKV in an endemic area due to travel, who were diagnosed and followed-up in an European country. The fact that the study was conducted in nine referral hospitals, most of them located in the two biggest cities of Spain (Madrid and Barcelona), likely facilitated the recruitment of most ZIKV-infected pregnant women in Spain during the study period. In addition, the participation of a diverse range of health professionals demonstrates the need for a multidisciplinary approach to researching this newly emerged congenital infection. In this manner, this study capitalized on the screening program in Spain during the ZIKV epidemics in Latin America. 

The results show that the great majority of our participants had been born in ZIKV endemic countries (95.7%), mainly Latin American and Caribbean countries. It was found that 70% were asymptomatic, and 14.7% classified as confirmed ZIKV. Interestingly, the RT-PCR assay remained positive beyond the first fifteen days after arriving in Spain in 58% of women with confirmed ZIKV. This result is consistent with previous observations and likely to be associated with a prolonged viremia [20]. In addition, nearly 10% of the children born to mothers with confirmed or probable maternal ZIKV infection had neurologic and hearing adverse outcomes possibly associated with ZIKV exposure. This result supports published observations from a number of different countries such as United States [13], French Territories in Americas [11], or French Guiana [10], but it is much lower than the 42% observed in the Rio de Janeiro cohort [9]. Regarding audiological tests, 3.5% of children had abnormal results, which is a smaller proportion compared to the from a Brazilian cohort [21], but significant in relation to the results of a study in USA in which all ZIKV exposed infants of traveler pregnant women passed the test [22]. Moreover, this percentage is 7-fold higher compared to rates of hearing loss in the general child population of 5 per 1000 live births in Spain [23]. However, it has to be noted that without longer-term follow-up we cannot infer the proportion of these children who will definitively have audiological adverse outcomes, or rule out the possibility of delayed hearing problems in children with normal results to date. 

The prevalence of CZS was 2.1% overall, increasing to one in eight among children whose mothers had confirmed infection, and congenital microcephaly was 40-fold higher than the estimated rate of 0.1 per 1000 live-births for this malformation in Spain [24]. Despite evidence that birth defects, including OPZRO, can be detected regardless of trimester of infection, for the three CZS cases here maternal infection occurred in the first trimester, as reported in other studies [9,25,26]. For one of the three CZS cases, no maternal symptoms related to ZIKV infection were reported.

It is important to note that similar to other studies, this cohort faced important challenges with respect to ZIKV laboratory diagnosis, including the short period for the detection of the ZIKV RNA [6] and cross-reactivity in serological assays for ZIKV and DENV [25]. Similarly, this research faced the lack of a gold standard serological test to determine the exact onset of ZIKV infection and the ability to exclude other flavivirus infections such as DENV [9,10,11,12,13,14]. Our national referral laboratory ISCIII in Madrid lacked the infrastructure to be able to perform any PRNT for other flaviviruses. Other limitations included difficulties in following up all the screened pregnant women, precluding comparison of pregnancy outcomes in ZIKV-infected women and in an uninfected control group. We were also unable to confirm the maternal ZIKV diagnosis in most cases, giving a much lower sample compared to other cohorts [9,10,11]. Finally, although we were able to assess frequency of OPZRO among exposed children with respect to audiological and neurological adverse outcomes, we had insufficient information regarding circumstances around elective or spontaneous abortions to be able to classify these outcomes. 

## 4. Materials and Methods

This multicenter prospective observational cohort study was conducted in nine referral hospitals. Women who attended antenatal clinics located at Primary Health Centers for Reproductive and Sexual Health were referred to these hospitals because they were classified as having been at risk of ZIKV exposure and therefore screened for ZIKV, as they came from or travelled to endemic areas during the pregnancy or the 2 previous months or their sexual partners had visited endemic areas in the previous 6 months, and because they tested positive for ZIKV-IgG or ZIKV-IgM. It is important to highlight that they were classified as exposed to ZIKV regardless the presence of symptoms. In addition, newborns were enrolled in the study at birth and followed for a maximum of 2 years. Data collected from the cohort of mother–child pairs was stored using REDCap Research Electronic Data Capture) [27]. The study obtained approval from the ethical committees of all participating centers and the study protocol followed the Spanish consensus document for the detection of ZIKV infection during pregnancy published at the Ministry of Health website [19]. 

### 4.1. Enrolment Criteria

Pregnant women were eligible for recruitment when transferred to referral hospitals for the reasons specified above. At recruitment, we collected demographic data, including maternal birth date, birth country, ZIKV exposure country, presence of symptoms, and departure and return dates from ZIKV endemic area. Likewise, blood and/or urine ZIKV samples were collected and analyzed via RT-PCR commercial assay (RealStar® Zika Virus RT-PCR kit 1.0, Altona Diagnostics), ZIKV-IgG, and ZIKV-IgM (IIFT Arboviral fever Mosaic IgG and IgM, Euroimmun, Germany from 2016 to December 2017; and ELISA Virus Zika IgG and IgM, Euroimmun, Germany, from December 2017 to the end of study) (Table 1). If ZIKV RT-PCR was negative but ZIKV-IgG tested positive, specimens were sent to the Instituto de Salud Carlos III (ISCIII, Madrid, Spain) for ZIKV-PRNT. However, ISCIII was not able to perform specific PRNT for DENV; consequently, positive or equivocal ZIKV-PRNT titers were not compared with DENV-PRNT results. Neutralization titer ≥1/32 was considered indicative of the presence of ZIKV neutralizing antibodies. The date of ZIKV test was considered to be the date of diagnosis for ZIKV infection. 

### 4.2. Cohort Follow-Up and Endpoints

Women were evaluated by the high-risk obstetrics team when confirmed or probable ZIKV infection was identified. Clinical evaluations, laboratory tests, and ultrasound (US) examinations were performed according to the established standard of care. Serological tests (IgG/IgM) for DENV and CHKV were used when possible to rule out co-infection with other arboviruses. Amniocentesis at 20–26 weeks of gestation with ZIKV RT-PCR testing of amniotic fluid was offered to all the mothers with probable or confirmed ZIKV infection. If any abnormality was identified during prenatal US examination, fetal MRI was performed. In addition, neonatal urine, serum, and saliva were sent for ZIKV testing. If microcephaly or other neurodevelopment abnormalities were detected at birth, diagnostic molecular tests for TORCH infections and a case history of participants were used to rule out other possible causes of microcephaly.

The study considered the following endpoints for pregnancy outcomes; delivery of a live-born infant with or without birth defects, miscarriage/spontaneous abortion, elective TOP before 20 weeks of gestation, and stillbirth (intrauterine fetal death at or after gestational age of 20 weeks or intrapartum death). The endpoints for children were the following clinical outcomes (by 31st March 2019); healthy or asymptomatic; CZS; OPZRO.

All children born to women with confirmed or probable ZIKV infection in pregnancy were followed-up to determine their clinical outcomes. According to standard of care, all of the participants were offered follow-up for their children at 1, 4, 9, 12, 18, and 24 months of age. This usually included neurological development assessment, ophthalmologic and hearing test, serological testing, cerebral US, MRI, and AABR test. Demographic, anthropometric, clinical, and laboratory data were collected for all the children born to ZIKV infected mothers. We defined an infant as a CZS case when she/he had any of the severe neurologic birth defects included in the currently proposed definition of CZS [4,17].

### 4.3. Statistical Analysis

Statistical analyses were conducted using Stata®v15 (Stata Corp LLC, Texas, USA) and descriptive statistics are presented for continuous and categorical data. To examine the association between maternal and infant characteristics and confirmed or probable maternal infection status, we used χ^2^ or Fisher exact tests according to sample size and reported *p*-values for each test. 

## 5. Conclusions

In conclusion, among children born to pregnant women with confirmed or probable ZIKV infection we found that nearly one in ten had neurologic and hearing adverse outcomes; where maternal ZIKV infection was confirmed, this risk increased slightly to 15.8%. All three of the CZS cases occurred when the mother was infected early in pregnancy. There is a need for continued and long-term follow-up of children born to infected mothers to assess the incidence and define the characteristics of any potential late-onset manifestations.

## Figures and Tables

**Figure 1 pathogens-09-00352-f001:**
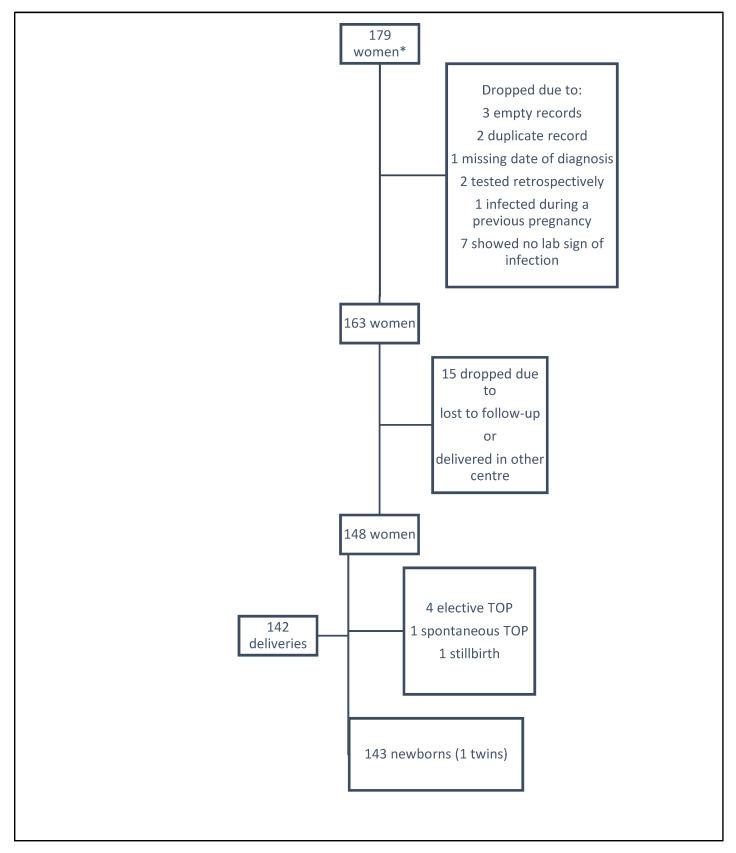
Flowchart of the study population of the mother–child pair Zika Spanish cohort. *These women were recruited from the total number of screened pregnant women in Spain due to their relationship to where they were being cared for their pregnancy. TOP: termination of pregnancy.

**Table 1 pathogens-09-00352-t001:** Classification of laboratory ZIKV diagnosis for pregnant women and their offspring.

	RT-PCR	IgG	IgM	PRNT
Confirmed	Positive			
	Negative or not done	Positive	Positive	Positive
Probable	Negative or not done	Positive	Negative	Positive
	Negative or not done	Positive	Positive	Negative
No evidence of Zika	Negative or not done	Positive	Negative	Negative
	Negative or not done	Negative	Negative	Not done

ZIKV: Zika virus; RT-PCR: reverse transcriptase polymerase chain reaction; IgG: immunoglobulin G; IgM: immunoglobulin M; PRNT: plaque reduction neutralization test for ZIKV.

**Table 2 pathogens-09-00352-t002:** Maternal and pregnancy characteristics by maternal Zika virus infection status.

	Confirmed Infection	Probable Infection	Total
	*n*=24	*n*=139	*n*=163
	n (%) or median [IQR]		
**Age at delivery (*n* = 157)**	29 (24–35)	27 (23–32)	28 (23–32)
**Total number of previous pregnancies (*n* = 162)**			
0	11 (46)	62 (45)	73 (45)
1	7 (29)	37 (27)	44 (27)
2	1 (4)	18 (13)	19 (12)
>=3	5 (21)	21 (15)	26 (16)
**Total number of previous live births (*n* = 162)**			
0	13 (54)	78 (57)	91 (56)
1	7 (29)	37 (27)	44 (27)
2	3 (13)	12 (9)	15 (9)
>=3	1 (4)	11 (8)	12 (7)
**Country of ZIKV exposure (*n*=163)**			
Dominican Republic	6 (25)	45 (32)	51 (31)
Honduras	4 (17)	34 (24)	38 (23)
Venezuela	0 (0)	17 (12)	17 (10)
Ecuador	3 (13)	11 (8)	14 (9)
Colombia	6 (25)	7 (5)	13 (8)
Bolivia	3 (13)	7 (5)	10 (6)
Other	2 (8)	18 (13)	20 (12)
**Gestational age at screening (completed weeks) (*n* = 162)**	19 (11–25)	22 (15–31)	21 (14–30)
**Clinical signs or symptoms (*n* = 161)**			
No	4 (17)	108 (78)	112 (70)
Yes*	20 (83)	29 (21)	49 (30)
Unknown	0 (0)	2 (1)	2 (1)
**Fever (*n* = 49)**	13 (68)	18 (67)	31 (67)
**Rash (*n* = 49)**	17 (85)	21 (72)	38 (78)
**Arthralgia (*n* = 49)**	12 (63)	16 (64)	28 (64)
**Conjunctival hyperaemia (*n* = 49)**	4 (22)	6 (24)	10 (23)
**Guillain–Barre syndrome (*n* = 49)**	1 (6)	1 (4)	2 (4)
**Focal neurological signs (*n* = 49)**	1 (6)	1 (4)	2 (4)
**Other signs and symptoms (*n* = 49)**	6 (33)	6 (23)	12 (27)
**Screening sample <15 days after arrival in Spain from endemic area (*n* = 141)**			
No	14 (58)	104 (89)	118 (84)
Yes	10 (42)	13 (11)	23 (16)
**ZIKV IgM (*n* = 163)**			
Negative	7 (29)	139 (100)	146 (90)
Positive	17 (71)	0 (0)	17 (10)
**ZIKV IgG (*n* = 163)**			
Positive	24 (100)	139 (100)	163 (100)
**Amniocentesis performed (*n* = 159)**			
No	8 (33)	121 (90)	129 (81)
Yes	16 (67)	14 (10)	30 (19)

* *p* < 0.0001 for symptoms between confirmed and probable infections.

**Table 3 pathogens-09-00352-t003:** Birth and infant outcomes (livebirth and stillbirths only), by maternal Zika virus infection status.

	Confirmed Infection	Probable Infection	Total	*p*-Value
	n = 19	n = 124	n = 143	
	n (%) or Median [IQR]			
**Gestational age at delivery (completed weeks) (*n* = 143)**	39 (37–40)	39 (39–40)	39 (38–40)	
**Preterm/term delivery (completed weeks) (*n* = 143)**				0.305
<34 weeks (very preterm)	1 (5)	1 (1)	2 (1)	
34–36 weeks (moderate preterm)	1 (5)	7 (6)	8 (6)	
>=37 weeks (term)	17 (89)	116 (94)	133 (93)	
**Sex (*n* = 141)**				0.475
Female	13 (68)	73 (60)	86 (61)	
Male	6 (32)	49 (40)	55 (39)	
**Birth weight (g) (*n* = 140)**				0.598
1500–2499	0 (0)	7 (6)	7 (5)	
>=2500	17 (100)	116 (94)	133 (95)	
**Birth weight z-score (n = 138)**				0.004
>=0	5 (29)	82 (68)	87 (63)	
−2-<0	12 (71)	38 (31)	50 (36)	
<−2	0 (0)	1 (1)	1 (1)	
**Head circumference at birth z-score (*n* = 129)**				0.054
>=0	9 (60)	85 (75)	94 (73)	
<0 & >=−2	4 (27)	28 (25)	32 (25)	
<-2	2 (13)	1 (1)	3 (2)	
**Length at birth z-score (*n* = 127)**				0.514
>=0	8 (57)	76 (67)	84 (66)	
−2-<0	6 (43)	35 (31)	41 (32)	
<−2	0 (0)	2 (2)	2 (2)	
**Type of delivery (*n* = 141)**				1.000
Spontaneous	15 (79)	90 (74)	105 (74)	
Assisted	1 (5)	6 (5)	7 (5)	
Emergency Caesarean	0 (0)	5 (4)	5 (4)	
Elective Caesarean	3 (16)	21 (17)	24 (17)	
**Child’s 1 minute Apgar score (*n* = 131)**				0.014
=<3	0 (0)	1 (1)	1 (1)	
4–6	2 (13)	0 (0)	2 (2)	
=>7	14 (88)	114 (99)	128 (98)	
**Child’s 5 minute Apgar score (*n* = 131)**				0.122
4–6	1 (6)	0 (0)	1 (1)	
=>7	15 (94)	115 (100)	130 (99)	
**Congenital microcephaly (*n* = 131)**	3 (16)	2 (2)	5 (4)	0.021
**Cranial/facial disproportion (*n* = 131)**	3 (16)	0 (0)	3 (2)	0.002
**Arthrogryposis (*n* = 130)**	1 (5)	0 (0)	1 (1)	0.146
**Biparietal depression (*n* = 131)**	2 (11)	1 (1)	3 (2)	0.009
**Excess nuchal skin (*n* = 131)**	2 (11)	0 (0)	2 (2)	0.054
**Adverse outcomes related to Zika (n = 143)**				0.001
Congenital Zika Syndrome	3 (16)	0 (0)	3 (2)	
Other possible Zika-related outcomes	0 (0)	11 (9)	11 (8)	
Asymptomatic	16 (84)	113 (91)	129 (90)	

**Table 4 pathogens-09-00352-t004:** Abnormal/adverse outcomes in children born to ZIKV-infected mothers in the cohort.

Case	Mother’s Country of Exposure	Week of Gestation at Diagnosis	Maternal Symptom	Method Diagnosis for Mother	Type of Maternal Infection	Week of Gestation at First US	Abnormal Prenatal Findings	Week of Gestationat Birth	Abnormal Physical Findings at Birth	Abnormal Findings at Follow-up
CZS cases										
1	Colombia	11	Yes	PCR-ZIKV (+) in serum, urine and amniotic fluid	confirmed	12	Yes (compatible with CZS)	37	Yes (compatible with CZS): congenital microcephaly (−5.1 z-score), craniofacial disproportion, biparietal depression, excess nuchal skin, and neurological abnormalities	CZS (congenital microcephaly, collapsed skull, craniofacial disproportion, arthrogryposis, hyperexcitability, hyperreflexia, abnormal mobility) Abnormal cerebral MRI findings (microcephaly with cortical atrophy affecting frontal lobes with pachygyria pattern in both cerebral hemispheres, delayed myelination pattern, microcalcifications in parieto-occipital regions, global thinning of the corpus callosum, and moderate supratentorial dysmorphic ventriculomegaly)
2	Ecuador	13	No	PCR-ZIKV (+) in serum, and amniotic fluid	confirmed	13	Yes (compatible with CZS)	22	Stillbirth (necropsy pathological findings were compatible with CZS): microcephaly (168mm; reference for this gestational age is 196 +/- 13mm), cortical atrophy, ventriculomegaly, bilateral polymicrogyria, leptomeningeal glioneuronal heterotopia, frequent calcifications in both hemispheres	CZS (microcephaly, partially collapsed skull, cranio-facial disproportion, and arthrogryposis) PCR-ZIKV was (+) in brain, CSF, placenta, thyroid, trachea, heart, quadriceps muscle, and bone marrow
3	Brazil	12	Yes	PCR-ZIKV (+) in urine, IgG-ZIKV (+), and IgM-ZIKV (+)	confirmed	29	Yes (compatible with CZS)	38	Yes (compatible with CZS): congenital microcephaly (-3.5 z-score), craniofacial disproportion, biparietal depression, excess nuchal skin, and neurological abnormalities	CZS (congenital microcephaly, collapsed skull, craniofacial disproportion, hyperexcitability, hyperreflexia, abnormal mobility) Abnormal cerebral MRI findings (microcephaly with cortical atrophy affecting frontal lobes with pachygyria pattern in both cerebral hemispheres, abnormal migration neuronal pattern, calcifications in brain parenchyma, basal ganglia and periventricular regions, dysgenesis of the corpus callosum, cisterna magna enlargement, and ventriculomegaly)
**Hearing loss findings**										
4	Dominican Republic	14	Yes	IgG-ZIKV (+) plus PRNT-ZIKV (+)	probable	14	No	40	No	Mild bilateral HL (abnormal AABR test at 10-m-old) LTFU after 10-m-old
5*	Dominican Republic	10	No	IgG-ZIKV (+) plus PRNT-ZIKV (+)	probable	12	No	39	Disproportion moderate microcephaly (−2.1 z-score below mean for age and sex)	Mild right HL (abnormal AABR test at 5-m-old) No microcephaly during follow-up time
6	Honduras	29	No	IgG-ZIKV (+) plus PRNT-ZIKV (+)	probable	34	No	40	No	Mild left HL (abnormal AABR test at 15-m-old) Normal cerebral MRI Language impairment
7	Bolivia	30	Yes	IgG-ZIKV (+) plus PRNT-ZIKV (+)	probable	31	No	39	No	Mild left HL (abnormal AABR test at 14-m-old)
8	Nicaragua	10	No	IgG-ZIKV (+) plus PRNT-ZIKV (+)	probable	12	No	39	No	Mild right HL (abnormal AABR test at 12-m-old)
**Abnormal postnatal cerebral MRI or head ultrasonographic findings in non CZS cases**										
9	Colombia	27	No	IgG-ZIKV (+) plus PRNT-ZIKV (+)	probable	34	No	38	No	Postnatal U/S: Bilateral lenticulostriate vasculopathy Postnatal cerebral MRI: left anterior arachnoid cyst
10	Dominican Republic	15	Yes	IgG-ZIKV (+) plus PRNT-ZIKV (+)	probable	21	No	35	No	Postnatal U/S: Hyper echogenicity of the bilateral periventricular white matter LTFU at 12-m-old
11**	Dominican Republic	21	Yes	IgG-ZIKV (+) plus PRNT-ZIKV (+)	probable	22	Yes (intrauterine growth retardation)	34	No	Postnatal U/S: hemorrhage of bilateral germinal matrix Postnatal cerebral MRI: Ventriculomegaly with intraventricular hemorrhage, ependymal and cisternae siderosis, loss of volume of bilateral cerebral white matter and thinned corpus callosum
12	Ecuador	39	Yes	IgG-ZIKV (+) plus PRNT-ZIKV (+)	probable	34	No	39	No	Postnatal U/S: lenticulostriate vasculopathy in right basal ganglia
13***	Dominican Republic	18	No	IgG-ZIKV (+) plus PRNT-ZIKV (+)	probable	18	No	40	No	Progressive microcephaly and postnatal growth retardation Postnatal cerebral MRI: craniofacial disproportion
14****	Honduras	36	No	IgG-ZIKV (+) plus PRNT-ZIKV (+)	probable	38	No	40	Yes (complex seizures)	Postnatal U/S: asymmetry of the germinal matrix Postnatal cerebral MRI: acute cortical-subcortical parietal and occipital bilateral ischemic lesions (left > right) due to probable embolic cause

AABR: auditory automated brainstem response; CZS: congenital Zika syndrome; HL: hearing loss; IgG-ZIKV: immunoglobulin G against Zika virus; IgM-ZIKV: immunoglobulin M against Zika virus; LTFU: lost-to-follow-up; MRI: magnetic resonance imaging; PCR-ZIKV: polymerase chain reaction for Zika virus; PRNT-ZIKV: plaque reduction neutralization test for Zika virus; U/S: ultrasonography. * Case 5 had a moderate disproportionate microcephaly at birth but no microcephaly was observed during the follow-up time. ** Case 11 had ventriculomegaly, loss of bilateral cerebral white matter volume, and dysgenesis of the corpus callosum alongside with intrauterine growth retardation. *** Case 13 had a HC within normal parameters at birth but we observed a progressive microcephaly and postnatal growth retardation during the follow-up time of study. **** Case 14 had complex seizures just after birth, the postnatal MRI showed multifocal arterial ischemic strokes.

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
