# Peer review of "Clinical Outcomes of a Zika Virus Mother–Child Pair Cohort in Spain"

_pathogens, 2020, doi:10.3390/pathogens9050352_

Round 1
Reviewer 1 Report
Manuscript Review: pathogens-781067
Clinical Outcomes of a Zika Virus Mother-Child Pair Cohort in Spain
Soriano-Arandes, et. al.
Summary:
The manuscript presented by Soriano-Arandes, et. al. outlines a predominantly very well described and presented prospective observational cohort study of ZIKV-infected pregnant women travelers within Spain following the epidemic outbreak of that virus in Latin America. The important results are clearly presented and support the authors conclusions.
Overall Opinion:
Overall, my opinion is that this article is suitable for publication with only a few minor changes necessary (see below for specific comments).
In addition to the purely scientific and formatting points below, I also would like to raise a minor concern regarding the authors elaboration on perceived importance of the immigrant status of the majority of these pregnant women travelers. To be clear, it is absolutely scientifically valid to state the demographics of these patients, providing a thorough background to the cohort, and I would suggest keeping the statements in section 2.1 of the results (from line 81 onwards) exactly as is. However, it does not seem to be a primary scientific aim of this study to consider the importance of nationality on the outcome of ZIKV-associated pregnancy, nor does the study compare outcomes in Spanish nationals to those of immigrants. Thus, comments such as those from lines 70-76 and lines 201-202 move beyond the scope of the study, touching upon sociological issues not under investigation here. It is sufficient for the introduction and discussion to simply refer to ZIKV-confirmed or -probable travelers (without indicating nationality), with the results briefly stating demographics in the absence of any supposition. This would not detract from the manuscript at all and would prevent any potential reader from misinterpreting conclusions.
Comments:
- A brief formatting issue – it seems the Materials and Methods may have moved from a position preceding the results. Several acronyms such as OPZRO and AABR are consequently not explained until late in the text, making this confusing for the reader.
- The authors need to refer to Figure 1 and Table 1 at the start of the Results section (I would suggest within section 2.1). In mentioning these references, the authors should also make brief statements explaining what constitutes a ZIKV-confirmed vs ZIKV-probable case, and a statement as to the drop in cohort numbers from 179 to 163.
Author Response
Dear reviewer,
Thank you very much for your comments. You'll find remarked in bold our responses to your questions:
- The manuscript presented by Soriano-Arandes, et. al. outlines a predominantly very well described and presented prospective observational cohort study of ZIKV-infected pregnant women travelers within Spain following the epidemic outbreak of that virus in Latin America. The important results are clearly presented and support the authors conclusions. Overall, my opinion is that this article is suitable for publication with only a few minor changes necessary (see below for specific comments). Thank you very much for your opinion.
- In addition to the purely scientific and formatting points below, I also would like to raise a minor concern regarding the authors elaboration on perceived importance of the immigrant status of the majority of these pregnant women travelers. To be clear, it is absolutely scientifically valid to state the demographics of these patients, providing a thorough background to the cohort, and I would suggest keeping the statements in section 2.1 of the results (from line 81 onwards) exactly as is. The description of the cohort participants has been added into the section 2.1 of the results as suggested by the reviewer.
- However, it does not seem to be a primary scientific aim of this study to consider the importance of nationality on the outcome of ZIKV-associated pregnancy, nor does the study compare outcomes in Spanish nationals to those of immigrants. Thus, comments such as those from lines 70-76 and lines 201-202 move beyond the scope of the study, touching upon sociological issues not under investigation here. This paragraph has been deleted following the recommendations of the reviewer. However, our intention was never to consider immigrant women a specific population associated with ZIKV infection because of their nationality. On the contrary, our intention was to demonstrate the vulnerability of these women to advocate for them in order to achieve better Public Health policies for this population. We apologize if this was misunderstood.
- It is sufficient for the introduction and discussion to simply refer to ZIKV-confirmed or -probable travelers (without indicating nationality), with the results briefly stating demographics in the absence of any supposition. This would not detract from the manuscript at all and would prevent any potential reader from misinterpreting conclusions. Following your recommendations the pregnant women are described in the section 2.1 according to their classification for ZIKV infection as confirmed or probable. Therefore, some of the references has been moved after adding this description here.
- A brief formatting issue – it seems the Materials and Methods may have moved from a position preceding the results. Several acronyms such as OPZRO and AABR are consequently not explained until late in the text, making this confusing for the reader. Thank you very much for advising us about these formatting issues. These acronyms have been corrected and previously explained into the main text.
- The authors need to refer to Figure 1 and Table 1 at the start of the Results section (I would suggest within section 2.1). In mentioning these references, the authors should also make brief statements explaining what constitutes a ZIKV-confirmed vs ZIKV-probable case, and a statement as to the drop in cohort numbers from 179 to 163. These changes have been made as suggested by the reviewer.
- The English style has been supervised by a native English person (Claire Thorne) which is one of the authors of the manuscript.
Reviewer 2 Report
This is an interesting clinical study paper about prospective cohort study of traveler mothers with confirmed or probable ZIKV infection and their children. In this study, methodological part is unclear and very limited information including serological, molecular assessment for Zika screening something like. In particular, authors did not mention about Ethics statement including IRB (Institutional Review Board) permission something. Overall, authors’ needs to improve including data presentation and methodology for publication in Pathogens.
Author Response
Dear reviewer,
Thank you very much for reviewing this article. Our responses to your questions have been added in bold at the end of each bullet point:
- This is an interesting clinical study paper about prospective cohort study of traveler mothers with confirmed or probable ZIKV infection and their children. Thank you very much for your opinion.
- In this study, methodological part is unclear and very limited information including serological, molecular assessment for Zika screening something like. A serological and molecular tests used in this study have been added specifying the mark of the test.
- In particular, authors did not mention about Ethics statement including IRB (Institutional Review Board) permission something. The approval obtained from Ethics Committee of each of the participating centres is mentioned in the line 276 in the material and methods part of the manuscript.
- Overall, authors’ needs to improve including data presentation and methodology for publication in Pathogens. We've tried to improve the manuscript following your comments and the comments made from other reviewers.
Reviewer 3 Report
Since reemergence of Zika virus (ZIKV) in 2015, ZIKV infection has been recognized as a public health threat causing microcephaly and other neuronal abnormalities in fetus. However probably due to vector mosquito distribution, the ZIKV infection has not impact on Europe comparing to Asia and South America. Thus it is important how ZIKV affect public health in Europe, particularly by immigrants from endemic countries. Authors conducted almost whole-country cohort study in Spain to understand ZIKV infection in pregnant women and their babies. Although the study is very valuable, there are several points to be revised.
Major Comments
- Authors should provide primer sequences used for RT-PCR and/or appropriate references.
- For IgG/IgM analyzes of ZIKV, dengue and chikungunya virus, did authors use commercial kits or perform experiments prepared in house. Please clarify and provide detail (Kit name used and/or methods).
- Have authors checked if the new born babies have anti-ZIKV antibodies ?
- Have authors confirmed that the stillbirth case is infected by ZIKV infection by pathological measures?
- Can authors provide the data showing the rates of microcephaly and hearing loss among new born baby in Spain before 2016?
Minor Comments
- Table 2 should not be appeared prior to Table 1.
- Line 97; DENV should be defined here.
- Line 99; CHIKV should be defined here.
Author Response
Dear reviewer,
Thank you very much for reviewing this article. Our responses to your questions have been added in bold at the end of each bullet point:
- Since reemergence of Zika virus (ZIKV) in 2015, ZIKV infection has been recognized as a public health threat causing microcephaly and other neuronal abnormalities in fetus. However probably due to vector mosquito distribution, the ZIKV infection has not impact on Europe comparing to Asia and South America. Thus it is important how ZIKV affect public health in Europe, particularly by immigrants from endemic countries. Authors conducted almost whole-country cohort study in Spain to understand ZIKV infection in pregnant women and their babies. Although the study is very valuable, there are several points to be revised. Thank you very much for your opinion, really very glad to hear about it.
- Authors should provide primer sequences used for RT-PCR and/or appropriate references. The RT-PCR for ZIKV that Spain used during this time was the RealStar® Zika Virus RT-PCR Kit 1.0, Altona Diagnostics. The description has been added to the methods part of the text.
- For IgG/IgM analyzes of ZIKV, dengue and chikungunya virus, did authors use commercial kits or perform experiments prepared in house. Please clarify and provide detail (Kit name used and/or methods). We used IIFT Arboviral fever Mosaic IgG and IgM, Euroimmun, Germany from 2016 to December 2017; and ELISA Virus Zika IgG and IgM, Euroimmun, Germany, from December 2017 to the end of study. This information has been added into the text. All the test used for other flaviviruses were commercial, not prepared in house.
- Have authors checked if the new born babies have anti-ZIKV antibodies? All the babies were checked for ZIKV antibodies, all of them were provided by serological tests for ZIKV at birth and at the following visits. An explanation has been included in the line 343.
- Have authors confirmed that the stillbirth case is infected by ZIKV infection by pathological measures? Yes, the stillbirth case was examined (a necropsy was performed) and pathological measures were compatible with CZS. We added a specific explanation in table 4 (case 2).
- Can authors provide the data showing the rates of microcephaly and hearing loss among new born baby in Spain before 2016?
- Regarding the congenital microcephaly we provide this information with a reference paper “Bermejo E, Cuevas L, Mendioroz J, et al. Frecuencia de Anomalías Congénitas en España: Vigilancia Epidemiológica en el ECEMC en el período 1980-2007. Revista de Dismorfología y Epidemiología. Serie V, Nº7, 2008, ISSN: 0210-3893”. The estimated rate of congenital microcephaly in Spain was from 0.2 per 1,000 live-birth in the period 1980-1985 to 0.1 per 1,000 live-birth in 2007.
- Regarding the congenital hearing loss we found a publication of 2003 talking about an estimation of a prevalence rate for this disease of 5 cases per 1,000 live-birth in Spain (Marco J and Mateu S. Libro Blanco sobre Hipoacusia.Detección Precoz de la Hipoacusia en Recién Nacidos, Ministerio de Sanidad y Consumo. NIPO: 351-03-007-8. Legal deposite: M-33935-2003.
- Both of these references have been added to the main text.
- Table 2 should not be appeared prior to Table 1. Agree, table 2 appear after table 1. It has been corrected.
- Line 97; DENV should be defined here. It has been amended.
- Line 99; CHIKV should be defined here. It has been amended.
Round 2
Reviewer 2 Report
I appreciate the care the authors took with revising the manuscript.
Author Response
Dear reviewer,
- I appreciate the care the authors took with revising the manuscript. Thank you very much for your comment.
Reviewer 3 Report
Authors essentially revised their manuscript.
One more point to be addressed.
1. Authors said that all new born babies were serologically tested against Zika virus in the responses. However, its results were not documented. Thus authors should mention if all babies associated with CZS were serologically positive or not.
Author Response
Dear reviewer,
Thank you very much for your comments.
Regarding to your following question:
1. Authors said that all new born babies were serologically tested against Zika virus in the responses. However, its results were not documented. Thus authors should mention if all babies associated with CZS were serologically positive or not.
The results you asked for are described in the lines 140-141 of the manuscript: "ZIKV-IgG was initially positive for the two live-born CZS cases but seroreversion was observed in both after the first 12 months of life."
The other CZS case is the stillbirth case that is described in table 4 (case 2). RT-PCR for ZIKV was (+) in brain, CSF, placenta, thyroid, trachea, heart, quadriceps muscle, and bone marrow. We did not collected samples for serological test.
Yours sincerely,